# Innovations in the Insect Cell Expression System for Industrial Recombinant Vaccine Antigen Production

**DOI:** 10.3390/vaccines9121504

**Published:** 2021-12-20

**Authors:** Manon M. J. Cox

**Affiliations:** NextWaveBio, LLC, East Haven, CT 06512, USA; manoncox@nextwavebio.com

**Keywords:** baculovirus, insect cell culture, protein production, antigen, vectors

## Abstract

The insect cell expression system has previously been proposed as the preferred biosecurity strategy for production of any vaccine, particularly for future influenza pandemic vaccines. The development and regulatory risk for new vaccine candidates is shortened as the platform is already in use for the manufacturing of the FDA-licensed seasonal recombinant influenza vaccine Flublok^®^. Large-scale production capacity is in place and could be used to produce other antigens as well. However, as demonstrated by the 2019 SARS-CoV-2 pandemic the insect cell expression system has limitations that need to be addressed to ensure that recombinant antigens will indeed play a role in combating future pandemics. The greatest challenge may be the ability to produce an adequate quantity of purified antigen in an accelerated manner. This review summarizes recent innovations in technology areas important for enhancing recombinant-protein production levels and shortening development timelines. Opportunities for increasing product concentrations through vector development, cell line engineering, or bioprocessing and for shortening timelines through standardization of manufacturing processes will be presented.

## 1. Introduction

The baculovirus–insect cell expression system has been extensively explored for production of viral antigens [1,2]. Viral antigens often require post-translation modifications and insect cells have the capability of performing many of those post-translational modifications such as glycosylation, disulfide bond formation, myristoylation, and phosphorylation [3]. The glycosylation profile is different from mammalian cells and glycoengineering has been successfully used to produce complex sialylated glycans and to eliminate α3 fucosylation either by generating transgenic cell lines or by using modified baculovirus vectors [4]. Veterinary vaccines manufactured using this production system to prevent classical swine fever, or to protect against porcine circovirus PCV-2 have been commercialized. Cervarix GSK’s (Rixensart, Belgium) bivalent human papilloma virus (HPV 16/18) vaccine against cervical cancer and Flublok, the first recombinant protein influenza vaccine developed by Protein Sciences Corporation (Meriden, CT, USA) are examples of FDA-approved human vaccines produced using this technology [5]. In addition to multiple efforts to develop a baculovirus-derived vaccine to prevent 2019 SARS-CoV-2 [6], there are numerous virus-like protein (VLP) vaccine products for other indications in development [7].

Baculoviruses (such as AcNPV (*Autographa californica* nuclear polyhedrosis virus)) have a double-stranded DNA genome of approximately 130 kb pairs in size that can easily be engineered to contain genes of interest as shown by Smith et al. nearly four decades ago when they expressed human β-interferon under the control of the polyhedrin promoter in insect cells [8]. That paper marked the beginning of the baculovirus–insect cell expression era and since then thousands of proteins have been produced using the polyhedrin promoter or, later, the p10 promoter to drive expression [1,2,5,6,7].

Here we focus on the development of a recombinant baculovirus through homologous recombination as this method has been shown to be useful for industrial-scale production [9]. Coding sequences from a foreign gene are inserted into a plasmid known as a baculovirus transfer plasmid using standard cloning techniques. The transfer plasmid contains the polyhedrin promoter upstream of a multiple cloning site, coupled to sequences naturally flanking the polyhedrin locus in AcNPV and a portion of the essential gene ORF1629 located downstream of the polyhedrin locus. ORF1629 encodes a phosphoprotein that is a key component of the nucleocapsid and generally accepted as essential to viral replication [10]. The transfer plasmid is co-transfected with baculovirus genomic DNA that has been linearized with an enzyme that removes the polyhedrin gene and part of ORF1629, rendering the non-recombined baculovirus DNA non-infectious [10]. Homologous recombination between the transfer plasmid and the linearized genomic DNA rescues the virus. The efficiency of recovery of recombinant viruses approaches 100% and plaques are nearly homogeneous, eliminating the need for multiple rounds of plaque purification. It is critical to keep virus propagation to a minimum as defective virus particles are formed in the process rendering the amplified baculovirus unstable [11,12]. The other broadly used laboratory method to generate recombinant baculovirus through site-specific transposition of Tn7 transposon into a bacmid containing the baculoviral genome (Bac-to-Bac Expression System (Invitrogen, Inc., Waltham, MA, USA)) has limited industrial use due to its inherent instability during scale-up [13].

Recombinant viruses can be propagated in cell lines derived from, for example, the fall armyworm *Spodoptera frugiperda* (SF) or the cabbage looper *Trichoplusia ni* (*T. ni*) [14], both of which grow well in suspension cultures [15]. Cell growth is arrested immediately post-infection as the virus transforms its host into a baculovirus and protein production facility.

The transient nature of the baculovirus–insect cell platform offers an important advantage as a single well characterized cell line is used for the production of any protein, thereby eliminating the time-consuming process of preparing, qualifying, and securing regulatory approval of a new cell line for each new protein. The disadvantage of the lytic infection process is that the production cycle is limited to the life-span of the cells and therefore reduces opportunities for cell-line improvement and fermentation optimization.

Fundamental research and major technological advances accomplished over three decades since the initial use of the baculovirus–insect cell system was extensively reviewed by van Oers et al. [5]. The authors identified minimizing the baculovirus genome as an important area for further improvement of the baculovirus vectors. 

The industrial production of proteins made in insect cells is the subject of reviews by Ikonomou et al. [16] and Roldão et al. [17] with an emphasis on cell culture and media development. There remains tremendous potential for improving yields obtained using this production system if we use the 200-fold improvement in productivity reported for monoclonal antibody in CHO cells as a reference [18] with today’s product concentrations exceeding 10 g/L [19]. The development of chemically defined media for insect cell culture is promising and the exploration of additives may further enhance productivity as previously shown for mammalian production processing.

Silencing of cellular and viral genes in the insect cell expression system using RNA interference technology [20] and adaptive laboratory evolution [21,22] are promising techniques to improve productivity either through modulation of the baculovirus vector or the cell line. Other genetic engineering strategies such as zinc finger nucleases (ZFNs), transcription activator-like effector nucleases (TALENs), and clustered regularly interspaced short palindromic repeats (CRISPR)/CRISPR-associated protein-9 (Cas9) that have been applied successfully in CHO cell glycoengineering applications [23] have not yet been extensively explored in the baculovirus insect cell system and are therefore outside the scope of this review.

This review summarizes recent innovations in technology areas important for enhancing recombinant protein production levels using the baculovirus–insect cell expression system and shortening development timelines. Opportunities for increasing product concentrations either by improving the productivity per cell or biomass concentration, and for shortening timelines through standardization of manufacturing processes will be discussed.

## 2. Baculovirus AcNPV Backbone and Transfer Vector Development

The baculovirus AcNPV backbone and the transfer vector have been genetically engineered over the years to improve ease of cloning and productivity. Widely used Ac-NPV backbone DNAs such as like *flash*BAC™ (Oxford Expression Technologies, Oxford, UK) and BaculoGold™ (BD Biosciences, Franklin Lakes, NJ, USA) are (linearized) engineered baculoviral DNAs containing a lethal deletion, as described earlier. Co-transfection of such backbones with a complementary baculovirus transfer vector restores viability by homologous recombination and rescues the virus along with the desired recombinant gene as described above [10]. The Oxford Expression System has commercialized two additional AcNPV backbones, *flash*BAC GOLD and *flash*BAC ULTRA [24,25] that combine the ease of cloning of the Bacmid system with homologous recombination [7]. In the Gold backbone the *chi*A (chitinase) gene and the v-*cath* (cathepsin) gene have been deleted. Removal of the *chi*A gene improves the efficiency of the secretory pathway and the v-*cath* reduces the chance of the recombinant protein being degraded although the latter can also be mitigated by adding leupeptin to the insect cells culture medium. In the ultra-version three more virus genes (p10, p74, and p26) have been eliminated from the *flash*BAC ULTRA genome. The absence of these genes removes an unnecessary genetic burden from the recombinant virus genome, providing a more efficient baculovirus expression vector. Specifically, the deletion of p10 increases polyhedrin promoter activity. Performance of the gold and ultra was compared in a benchmark study and showed superior performance for the Ultra vector [26]. Interestingly, Bacmid-like vectors outperformed the recombinant baculoviruses substantially, but scalability of these vectors has been problematic due to the instability of the viruses [13] and as such the industrial use is limited as previously noted.

Van Oers et al. [5] postulate that an estimated 40 genes of the AcNPV genome could be eliminated in the context of cultured cells leaving a lot of room for further development of the backbone. 

RNAi technology has been applied to the AcNPV backbone with the goal being to improve production of recombinant proteins [20]. The v-*cath* gene was used as a model gene and Kim et al. [27] confirmed that infected cells in which the gene was silenced exhibited higher viability, reduced proteolysis, and up to a three-fold increase in recombinant GFP compared to control cells. Furthermore, a limited silencing of gp64 using siRNA [28] or dsRNA [29] has proven to reduce residual baculovirus contaminants and increase the yield of a recombinant protein by 30%. The reduction of residual baculovirus contaminants has the added benefit of simplifying the downstream purification process (see also below). Salem et al. [30] demonstrated that silencing of ORF34, a transcriptional unit without a known function, but essential for baculovirus spread, enhances heterologous gene expression and Zhang et al. [31] used short hairpin RNA (shRNA) expression cassettes targeting a conserved region in SF caspase-1 and T.ni caspase-1 resulting in suppressed cell apoptosis and superior recombinant protein productivity. 

Algenex (Madrid, Spain) developed an expression cassette named TB consisting of a cDNA encoding the baculovirus transactivation factors IE1 and IE0, expressed under the control of the *pol*h promoter, and a homologous repeated transcription enhancer sequence operatively cis-linked to p10 chimeric promoter. This vector resulted in significant production improvements, including prolonged cell integrity after infection, improved protein integrity, and up to a four-fold increase in recombinant protein production yields in insect cells [32,33].

Paratechs Corporation (Lexington, KS, USA) developed a vector that includes a vankyrin gene that delays cell lysis and increases recombinant glycoprotein yield [34]

AB Vector (San Diego, CA, USA) offers a range of Profold transfer vectors that contain human molecular chaperones. ProFold™-C1 vector provides human Hsp40 and Hsc70, major cytoplasmic molecular chaperones, at levels comparable to levels of synthesis of a target protein. Similarly, ProFold™-ER1 provides major endoplasmic reticulum molecular chaperones that facilitate folding of target proteins in the endoplasmic reticulum. ProFold™-PDI provides protein disulfide isomerase (PDI) to facilitate folding of cysteine-rich proteins. The transfer vectors are generally compatible between various suppliers. 

Overall, commercially available transfer vectors offer a limited choice in promoter (s), signal sequence, purification tags. Table 1 highlights recent main commercially available transfer vectors.

Unfortunately, there are no comprehensive benchmark studies comparing the performance of vectors from different suppliers. 

Other features to improve gene expression through promoter optimization, combination of promoters, and/or use of enhancers have been described in the literature and were recently reviewed by Grose et al. [35]. For example, target protein production can be improved when duplicating the burst sequence compared to the polyhedrin promoter alone [36] and inserting the enhancer homologous region 1 (HR1) increased polyhedrin expression by 11-fold when placed downstream from the luciferase reporter gene [37]. More recently, Gwak et al. [38] described a vector that showed approximately 94-times greater and one-day-earlier expression of the foreign protein than the control vector containing only the polyhedrin promoter. The factors that increased the expression efficiency of the polyhedrin promoter were the repeated burst sequences, the p6.9 promoter, and hr3.

Lou et al. [39] explored three additional signal sequences besides gp64 and the honeybee signal sequence when expressing human thyroid peroxidase in *T. ni* cells, i.e., ecdysteroid UDP-glucosyl transferase (EGT), human peptidyl-glycine alpha-amidating monooxygenase (PAM), and human azurocidin. Interestingly the PAM signal peptide enhanced the hTPO secretion about 2.5-fold.

Co-expression of insect initiation translation factors in addition to the use of chaperones is a way to improve expression levels even further as demonstrated by Teng et al. [40] when co-expressing insect translation initiation factor eIF4E with human chaperones calreticulin (CALR) or β-synuclein (β-syn). The production of a recombinant secreted alkaline phosphatase (SEFP) increased substantially in comparison to using chaperones alone.

Combining described approaches beneficial for increased productivity within a defined baculovirus transfer vector and genomic baculovirus backbone could result in substantially higher yields, thereby simplifying purification as the ratio of product of interest to contaminants will improve as well. 

The challenge, of course remains in translating the findings into a standardized vector that works well for multiple different products of interest.

## 3. Cell Line Engineering

Cellular engineering is a promising methodology to improve recombinant protein production. Relatively little work has been done to engineer insect cell lines to improve productivity, most likely due to the fact that the infection process is lytic and cells die during the production process. 

Recent work has shown great promise for adaptive laboratory evolution (ALE) of cell lines. The use of ALE, i.e., adaptation of cells to efficiently grow under non-standard culture conditions, through consecutive sub-culturing under these selective pressures allows for the selection of cell populations with enhanced fitness. ALE has been suggested as an approach to maximize recombinant protein titers in both prokaryotes and animal cells [41,42]. 

The first indication for possible success in insect cells came from work by Wagner et al. [43] who showed that a novel insect cell variant derived by exposure of SF to elevated culture pH for a prolonged period of time was capable of maintaining normal cell growth into the typical mammalian cell culture pH range of 7.0–7.2 and produced 11-fold higher Chikungunya VLP yields compared to the parental SF cell line. 

Correira et al. [22] used a similar approach to adapt *T. ni* insect cells to grow at a neutral culture pH (7.0) resulting in improved production of influenza hemagglutinin (HA)-displaying virus-like particles (VLPs). The cell-specific HA productivity was increased three-fold and volumetric HA titer of up to four-fold as compared to non-adapted cells, whereas a pH shift alone did not improve yield. 

Fernandes et al. [21] used ALE to improve the production of HIV-Gag virus-like particles (VLPs) in stable SF and *T. ni* cell lines. Cells were cultured under controlled hypothermic conditions (22 °C instead of standard 27 °C) for a prolonged period of time (over 3 months), which allowed the selection of a population of cells with an improved phenotype. Adapted cells expressed up to 26-fold (SF cells) and 10-fold (*T. ni* cells) more Gag VLPs than non-adapted cells cultured at standard conditions. Evaluation of the performance of these cell lines after transfection with a recombinant baculovirus in a lytic cell line setting remains to be done.

Bottlenecks in transcription, translation, protein processing, secretory pathways, and viability can be addressed in a more targeted manner using RNAi technology as our understanding of the host cell improves. For example, successful application of this technology resulted in improved production of recombinant proteins when delaying apoptosis by controlling expression of caspase-1 in cell lines and/or controlling the cycle by downregulating of cyclin E (a positive regulator of G1- to-S phase transition). Expression levels of GFP [44,45], SEAP [46], and the fusion protein Tim4-Fc [47] were 100%, 100%, and 400% higher than the parent cell lines following caspase-1 silencing. Wu et al. [48] demonstrated an almost two-fold increase in recombinant GFP expression when silencing of Cyclin E was induced briefly before baculovirus infection.

The disadvantage of improving productivity through cell line engineering may be that new or additional characterization of the cell line would be needed and this would substantially increase the timeline. Therefore, it will be necessary to evaluate which features are likely to improve performance for multiple different antigens such that they can all be combined in a single well-characterized cell line.

## 4. Insect Cell Culture

The baculovirus insect cell culture process has three components as shown in Figure 1: cell expansion, virus production, and protein production. Insect cells are generally sub-cultured when cells reach mid-log phase of growth, around 5 × 10^6^ cells mL^−1^, and are diluted ~10-fold into fresh medium during the cell expansion phase. Cells must be in their early log phase to support optimal infection which is around 1.5 × 10^6^ cells mL^−1^ under standard cell culture conditions. The addition of the virus results in an arrest in cell growth as the virus invades the host cell and takes over. New baculovirus particles are formed and protein production begins. Baculovirus stocks are generated by infecting cells with a multiplicity of infection (MOI) that is approximately 10-fold lower than the one used for protein production. Virus stocks are also harvested earlier (2 days post infection) to limit the formation of defective interfering particles. Protein production cultures are generally infected with an MOI of one and are harvested when protein yield is maximized (three or more days post infection). Conditions may be unique for each protein and, therefore, it is recommended to determine optimal conditions as described for the Zaire Ebola virus-like particles by Pastor et al. [49].

Insect cell culture conditions have been reviewed in detail by Roldão A et al. [17] and are briefly summarized here. The optimal temperature for insect cells cultivation is 27 °C [50] at which maximum growth rates (μ max) are between 0.029 and 0.039 h^−1^ (duplication time of 18–24 h) and cell densities vary from 0.6 to 1.8 g L^−1^ (cell dry weight). Note that this temperature may not be optimal for protein or virus production as shown by Fernandes et al. [21] and others.

The optimal pH of most insect cell cultures is around 6.0–6.4 [51] and typical osmolarities of insect cell cultures are between 300 and 380 mOsm L^−1^. Also, for pH there are examples that growing cells at elevated pH is beneficial for VLP or protein production [22,43].

The control of dissolved oxygen (DO) is essential to avoid oxygen limitation or excess, inhibiting the synthesis of proteases or oxidative damage to proteins [52], and maximizing cell growth.

Insect cell culture in bioreactors is generally performed in batch, or fed-batch mode, with the main challenge being finding a balance between optimal cell density and productivity per cell when infecting the cells with the recombinant virus.

The productivity of the cell culture process can be increased dramatically by increasing the cell mass prior to initiation of the infection process while maintaining the cells in their logarithmic growth phase. This can be accomplished by optimizing the feed formulation and the feeding strategy. For example, Meghrous et al. [53] described a 2–5-fold improvement in hemagglutinin production by implementation of a simple feed strategy and maintaining the cells for longer in their early logarithmic phase. Further improvement should be feasible as exemplified by the 40-fold improvement in antibody production described for CHO cells using the fed-batch process [54]. 

While most insect cell culture processes are operated with limited process controls it may be important to consider managing the CO_2_ concentration as it accumulates as a by-product of cellular growth and inhibits cell growth and protein expression [55,56]. 

Optimization of the cell-culture growth medium offers enormous opportunity for improvement of productivity per cell or biomass concentration but also for facilitating purification of the product of interest. 

A full understanding of the cell culture process and insect cell metabolism including the use of a chemically defined cell culture medium will be beneficial as the composition of the “wasted” medium is known and can be systematically addressed during the purification.

Knowledge of insect cell metabolism is necessary for the rational design of optimal cell culture medium but remains limited [16]. The composition of the complex medium required to grow animal cells is a key factor in ensuring consistent recombinant protein production from a producer cell. 

Media formulations can contain 60–100 components, which change in concentration during a batch culture [57]. Glucose and glutamine are key nutrients utilized for energy metabolism during cell growth. Glucose is considered the most important carbon source for insect cell growth [58,59]. Insect cells cannot synthesize most of the amino acids by themselves [60] and, therefore, supplementation of amino-acids to the medium is essential to guarantee cell growth and protein expression. Other nutritional requirements of critical importance for cellular growth and/or product formation are lipids, cholesterol, and vitamins. Because insect cells are unable to synthesize, desaturate, and elongate fatty acids, supplementation of culture media with lipids is essential to avoid cell degeneration and formation of defective interfering particles (non-infectious viruses) [61,62]. Supplementation of cholesterol and vitamins is also required (insect cells cannot produce them) as they play major roles in membrane formation and regulation of key metabolic enzymes [63]. Hence the cell culture media used for growth and protein production is complex and contains many ingredients including glucose, amino-acids, vitamins, trace elements, lipids, and yeastolate [54]. The non-ionic detergent Kolliphor^®^ P188 (also known as Lutrol^®^ F68, Pluronic^®^ F-68, Synperonic^®^ F68, or Poloxamer 188) is often added to the culture medium as a protective agent to prevent cell damage [64].

Protein-free chemically defined media and feeds have been reported for animal cells including CHO [54] and NS0 cells [65,66]. 

Recently, commercial insect cell media suppliers have focused on the development of chemically defined media free of animal components and not subject to the variability caused by undefined ingredients such a yeastolates. A list of commercially available media is shown in Table 2.

Few papers compare different commercial media in a comprehensive manner. One example provided for the ExpiSf media shows that significantly higher AAV yields were obtained by using the ExpiSf Baculovirus Expression System with a clinically applicable, CD culture medium. For the three serotypes tested, AAV9 yields increased ~7-fold, AAV2 yields increased ~15-fold, and AAV8 yields increased ~19-fold. This data demonstrate that the ExpiSf expression system allows facile and scalable manufacturing of AAV vectors in CD medium [67].

Additives that increased recombinant protein yields in CHO cells, recently reviewed in detail by Li et al. [68], may also improve yields in insect cells. Briefly, the authors divide small molecule substances that stimulate the expression of antibodies into two categories: (1) carboxylic acids, of which sodium butyrate (NaB) is the most typical representative; and (2) antioxidants such as ascorbic acid and reduced glutathione. Sodium butyrate is an inhibitor of histone deacetylation, and possibly increases gene transcription by enhancing gene accessibility to transcription factors [69].

An example from CHO cell production shows that the potential for yield improvement can be dramatic: The addition of 3mM NaB in combination with overexpression of the anti-apoptotic protein bcl-2 resulted in a ten-fold increase in the concentration of recombinant human thrombpoietin (hTPO). Both the enhanced specific productivity induced by NaB and the extended culture longevity provided by the antiapoptotic effect of Bcl-2 overexpression contributed to the enhancement of maximum hTPO concentration. [70]. Another example is that the expression rate of the tissue-type plasminogen activator was increased by adding antioxidants (ascorbic acid and reduced glutathione) to the CHO cell culture process [71]. 

NaB and DMSO had a positive impact on growth kinetics and P24 production in insect cells similar to achievements described for other cell lines [21]. The expression of recombinant proteins following transduction of CHO cells with recombinant baculoviruses containing a mammalian expression cassette with the CMV-promoter was enhanced by the addition of trichostatin A (TSA), a specific histone deacetylase inhibitor [72]. 

## 5. Downstream Processing

During the purification process, the baculovirus–insect cell system contaminants are removed from the product of interest. Full characterization of process contaminants would be beneficial for establishing the most efficient purification process. A lot of work remains to be done here but, roughly, contaminants can be divided in the following categories: host cell DNA, baculovirus, host cell proteins, baculoviral proteins, media residuals, and waste products. 

A universal protein purification process would be designed around the removal of these contaminants and once established could dramatically reduce development timelines. An example of a universal purification process has been described for recombinant hemagglutinin produced in insect cells [9]. Briefly, infected cells (containing rHA) are removed from the protein production bioreactor(s) and separated from the culture media by centrifugation. The cell pellet containing insect cell membranes is solubilized in buffer containing a non-ionic detergent and the extract is subsequently clarified using depth filtration. The rHA-containing extract is subsequently applied to an ion-exchange column. The buffer pH is selected such that rHA proteins will bind to the cation exchange column. DNA contaminants will bind to the anion column. rHA is eluted and then applied to a hydrophobic interaction column where it binds strongly whilst most protein contaminants flow through. The rHA-containing eluate is then applied to a Q-membrane to remove residual DNA and finally ultrafiltration is used to formulate the protein in its final buffer. 

In the event the protein or product of interest is secreted into the medium, the key will be to separate the contaminants present in the medium from the product of interest. Generally, the first step in such a purification scheme includes concentration of the medium to reduce processing volume. Subsequently the product of interest can be captured using affinity chromatography and/or ion-exchange chromatography. The main challenge when purifying a secreted product is presented by the recombinant baculovirus present in the medium that often has similar biochemical characteristics to the product of interest. This could possibly be addressed through genetic engineering as described below.

Cell-line engineering or baculovirus-vector engineering can take purification one step further. For example, to avoid the production of contaminating baculovirus, the use of a Δvp80 baculovirus backbone for the production of a recombinant protein was evaluated [73]. Vp80 encodes the nucleocapsid-associated glycoprotein VP80. Knock-out of this protein completely blocks production of baculovirus particles without reducing transgene expression. 

As described earlier, limited silencing of gp64 using siRNA [28] or dsRNA [29] reduces residual baculovirus contaminants and increased the yield of a recombinant protein 30% hence simplifying the purification process.

Knocking out many of the other estimated 40 genes unnecessary for the production of the recombinant protein in insect cells would clearly benefit the purification process of any product of interest [5].

The use of enzymes such as DNase for DNA removal has not been broadly explored but could offer major benefits as they are routinely used in other industrial production processes.

## 6. Conclusions

In this review we presented multiple strategies to further improve productivity either through vector development, cell line engineering, cell culture, and/or purification. Successful strategies have been aimed at increasing transcription, increasing translation, preventing proteolysis, extending the life cycle of the cells (modulating apoptosis), arresting cells in their growth cycle, or simply increasing the biomass while keeping the cells susceptible for optimal protein expression.

Silencing of cellular and viral genes in the insect cell expression system using RNA interference technology [20] and adaptive laboratory evolution [2,21] are promising techniques to improve productivity through modulation of either the baculovirus vector or the cell line. Multiple examples were presented.

In addition, progress has been made in developing a chemically defined medium and various additives that were previously shown to be effective in mammalian cell culture are also beneficial in improving productivity in insect cells. Other bioprocessing learnings from recombinant protein or monoclonal antibody production in CHO cells or *E. coli* may also be applicable to insect cell culture. 

A systematic evaluation of the performance of baculovirus transfer vectors, the backbones and cell culture media for various of proteins would be beneficial and facilitate the selection of a system for optimal production of target proteins.

Standardization of production is critical to reduce development timelines as demonstrated for the universal manufacturing process of the first recombinant influenza vaccine approved by the FDA as described in the biological license application (BLA). This allowed for production of a purified protein antigen (or vaccine component) at the 10 L scale accomplished within 38 days under GMP conditions with the same process performance at the 2 L, 10 L, 100 L, 650 L, and 2500 L scale [74].

## Figures and Tables

**Figure 1 vaccines-09-01504-f001:**
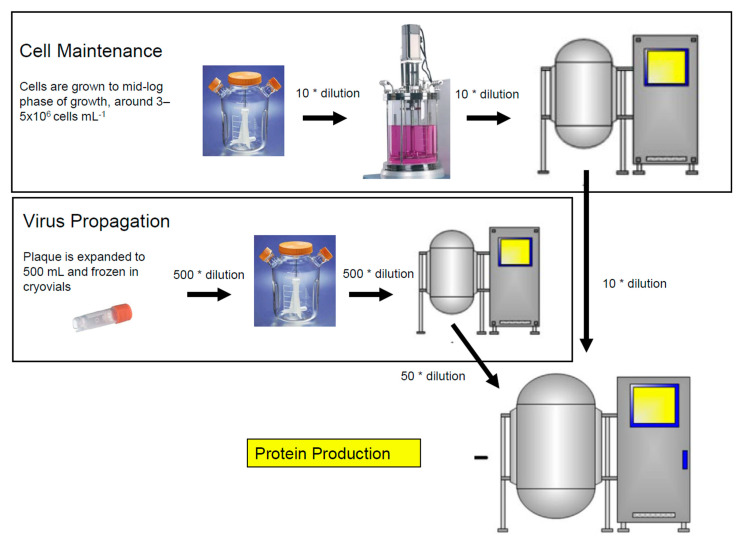
Three stages of the baculovirus insect cell culture process. Cell maintenance, virus propagation, and protein production.

**Table 1 vaccines-09-01504-t001:** Commercial transfer vector features.

Features	Description	Supplier (Assessed on 19 December 2021)
Promoter	Polyhedrin, P10, or basic promoter (late promoter) (single or double copy)	BD Biosciences (www.bdbiosciences.com)
Promoter + transactivation factors	Polyhedrin promoter plus transactivation factors IE1 and IE0 + HR linked to p10 chimeric promoter	Algenex (www.algenex.com)
Signal sequences	Acidic glycoprotein gp67 (also known as gp64)	BD Biosciences (www.bdbiosciences.com)
	Honeybee	ThermoFisher (Waltham, MA, USA) (www.thermofisher.com)
Delay cell death	Baculovirus vectors encoding P-vank-1 gene.	ParaTechs (www.paratechs.com)
Purification tags	His-tag or GST tag	BD Biosciences
Chaperones	Hsp40 and Hsc70 major ER molecular chaperones/disulfide isomerase	AB Vector (www.abvector.com)

**Table 2 vaccines-09-01504-t002:** A summary of suppliers, media, and their main characteristics.

Supplier Website (Assessed on 15 October 2021)	Medium Name	Chemically Defined	Animal-Component Free	Hydrolysate-Free	Serum-Free	Protein-free	Reported Max Cell Density Cells/mL
www.thermofisher.com	Sf-900 III (SFM)	N	Y	N	Y	N	1–1.4 × 10^7^
	ExpiSf CD	Y	Y	Y*	Y	Y	2 × 10^7^
www.sartorius.com	4Cell^®^ Insect CD	Y	Y	Y	Y	Y	1 × 10^7^
www.expressionsystems.com	ESF AF contains L-Glutamine and Pluronic^®^ F-68	N	Y	ND	Y	Y	2 × 10^7^
www.fishersci.com	Insectagro™ with L-Glutamine	N	ND	ND	Y	Y	ND
www.cytivalifesciences.com	SFM4Insect™ contains L-Glutamine and poloxamer 188	N	Y	ND	Y	Y	ND
www.bdbiosciences.com	BaculoGold Max-XP	N	ND	N	Y	N	ND
www.labchem-wako.fujifilm.com	WakoVAC PSFM-J1	N	ND	N	Y	N	ND

N = No; Y = Yes; * = low hydrolysate; ND = Not disclosed.

## Data Availability

Not applicable.

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
