# Peer review of "Innovations in the Insect Cell Expression System for Industrial Recombinant Vaccine Antigen Production"

_vaccines, 2021, doi:10.3390/vaccines9121504_

Round 1
Reviewer 1 Report
The author of the review describes the baculovirus/insect cell system as an interesting alternative to produce recombinant vaccine antigen ranging from transfer-vector design, cell line engineering to downstream bioprocessing with a focus on yield improvements.
General comments:
- In section 1 the author describes different baculovirus backbones and transfer-vectors with a strong focus on homologous recombination-based systems. The author only briefly mentions that bacmid based systems are available as well, but industrial use might be limited due to their lower genetic stability (are their additional references available). Nevertheless, readers might be interested in these vectors as well because there are interesting systems available allowing for the co-expression of several different genes from one viral backbone or performing virus based cellular engineering.
- In section 4 (Downstream processing) the author describes different cell line / virus engineering efforts to reduce process related impurities but is not mentioning any technological possibilities for improvement in the fields of chromatography etc. Maybe the author can extend that section because downstream processing plays a crucial role especially when it comes to the production of virus-like particles.
- I would ask the author to re-work the figures because line-art is often not appropriate.
Specific comments:
- Line 25: The author mentions that insect cells can perform glycosylation. Although this is true pre se patterns greatly differ from mammalian structures. This can lead to the fact the glycoengineering might be necessary to express the antigens correctly.
- Line 33: Please rephrase: “….in size- or AcNPV ()- that can easily…) because it is misleading.
- Line 42: What is the essential gene ORF1629 for?
- Line 142: Readers might be interested in the names of the commercial transfer vectors.
- Line 221: Please rephrase: “Insect cells grown are generally…..”
- Line 231: MOI of one for protein production. Is this always the case?
- Line 235: The figure shows a 50 fold dilution of the virus for infecting the main culture. Du to the fact the MOI is based on the cell count this is misleading.
- Line 288: Pluronic F-68 is currently mostly marketed under the name Kolliphor P188.
- Line 304: Table 2 (Line 1 is not readable).
- Line 317: NaB or NaBu?
- Liner 339: The is no graphic for extraction in figure 3.
Author Response
Thank you for these constructive comments.
Please find attached my responses.

Reviewer 2 Report
The article 'Innovations in the insect cell expression system for industrial recombinant vaccine antigen production'. The article is well written and highlights the role of insect cell expression system. However, the author could consider the following comments to improve the article.
- A brief history of the insect cell expression system is required in the introduction.
- The article needs to include more information related to the state of art recombination mechanisms available to date.
- Can Zinc finger nucleases or TALENS for the use in the insect cell expression system. The authors need to further explore such gene editing mechanisms related to this topic, a separate section included in the review is beneficial.
- The author mentioned only one product that is approved by the FDA, any information about the upcoming targets that are considered for insect cell expression and production is beneficial to understand the scope of the field.
- What is the potential market size of various expression systems such as lentiviral, AAV and insect cell expression. How does this system compare with others. a brief chart describing it would be beneficial.
- Can the author summarize the methods that are involved downstream purification systems.
Author Response
Thank you for your constructive comments.
Please find attached my responses.
Kind regards, Manon Cox

Reviewer 3 Report
The manuscript submitted by Manon M.J. Cox, entitled Innovations in the insect cell expression system for industrial recombinant vaccine antigen production aims to review the advantages and disadvantages of the insect cell expression system to produce vaccine antigens, taking in consideration also the COVID-19 pandemics.
In the opinion of this reviewer, many references are missing and the manuscript doesn't bring a significant contribution to the field. Thus, this work should not be accepted for publication in a journal like Vaccines.
Author Response
This review paper focuses on which innovations in the insect cell expression system for industrial scale recombinant antigen production are relevant such that it may be relevant to combat future pandemics. The author is the lead developer of Flublok the first recombinant FDA approved influenza vaccine produced in insect cells. As such she has first hand experience on what is needed to ensure that this technology can be successfully applied in future pandemics. The core reason why there is no commercial SARS-CoV2 vaccine made in insect cells available to date relates to product concentrations and production timelines and require a comprehensive multi-disciplinary approach.
Recognizing that no single discipline (no matter how brilliant) can lead to a commercial product is key to understanding why this review is important.
Round 2
Reviewer 1 Report
The author very well addressed all points mentioned and readers will definitely profit form the update. Thank you!
There are two typos that still need to be addressed:
- Line 44: "Here we focus on the of a ....."
- Line 353: "...recombinant hemagglutinin produced insect cells".
Author Response
Thank you for these final comments.
The letter to Editor has been uploaded to show how comments were addressed.

Reviewer 3 Report
The author had improve the quality of the manuscript following all the suggestions of the reviewers.
Author Response
Dear Reviewer,
Thank you for your prompt review.
The letter to Editor has been included to show how comments were addressed.
Kind regards, Manon
